# Data Augmentation for Compositional Data: Advancing Predictive Models of the Microbiome

**Elliott Gordon-Rodriguez**
Columbia University
eg2912@columbia.edu

**Thomas P. Quinn**
Independent Scientist
contacttomquinn@gmail.com

**John P. Cunninghham**
Columbia University
jpc2181@columbia.edu

## Abstract

Data augmentation plays a key role in modern machine learning pipelines. While numerous augmentation strategies have been studied in the context of computer vision and natural language processing, less is known for other data modalities. Our work extends the success of data augmentation to *compositional data*, i.e., simplex-valued data, which is of particular interest in microbiology, geochemistry, and other applications. Drawing on key principles from compositional data analysis, such as the *Aitchison geometry of the simplex* and *subcompositions*, we define novel augmentation strategies for this data modality. Incorporating our data augmentations into standard supervised learning pipelines results in consistent performance gains across a wide range of standard benchmark datasets. In particular, we set a new state-of-the-art for key disease prediction tasks including colorectal cancer, type 2 diabetes, and Crohn's disease. In addition, our data augmentations enable us to define a novel contrastive learning model, which improves on previous representation learning approaches for microbiome compositional data.[1]

## 1 Introduction

Data augmentation, i.e., generating synthetic training examples to be used in model fitting, is a core component of modern deep learning pipelines [54, 17]. In computer vision, augmentations such as image resizing and shifting have been used since as early as LeNet-5 [35]. These and many other augmentations have become essential to highly successful state-of-the-art architectures, ranging from AlexNet [34] and ResNet [29] to recent contrastive learning models such as SimCLR [9] and MoCo [28]. As such, a growing body of literature has emerged to develop and characterize data augmentation techniques, particularly in computer vision [45, 66, 13, 10, 11, 65], as well as natural language processing [53, 12, 17]. However, defining useful data augmentations is highly domain-dependent, and fewer works have studied augmentations for more general data modalities, such as tabular data [66, 64]. The goal of our work is to extend the success of data augmentation to a previously unexplored data modality; namely, compositional data.

Compositional data (CoDa) are those that represent the parts of a whole, and therefore carry only *relative* information. Equivalently, we can think of CoDa as a set of datapoints living in the simplex:

$$\mathcal{D} = \left\{ \boldsymbol{x}_i \in \Delta^{p-1} \right\}_{i=1}^{n}, \quad \text{where} \quad \Delta^{p-1} = \left\{ \boldsymbol{x} \in \mathbb{R}_+^p : \sum_{j=1}^{p} x_j = 1 \right\}. \tag{1}$$

Examples of CoDa arise throughout the sciences, including microbiology (proportions of different species in a bacterial community) [23, 22, 50, 3], geochemistry (chemical composition of a geological sample) [5, 4, 42], materials science (components of matter) [40, 1], and many more. As a result,

---

[1]Our code is available at `https://github.com/cunningham-lab/AugCoDa`.

36th Conference on Neural Information Processing Systems (NeurIPS 2022).

starting with the seminal work of Aitchison [1], numerous bespoke techniques have been developed for the statistical analysis of CoDa [44, 43, 18].

A rapidly growing area of application for CoDa is the human microbiome, which comprises the populations of microorganisms that reside inside and on the human body [59]. Microbiome data arise from an inexhaustive sampling procedure as a result of high-throughput sequencing [23, 51]. In particular, each feature typically represents the *relative abundance* of some species of microorganism; as such, each observation can be normalized to the simplex prior to downstream analyses [23, 22, 50, 37, 41].

The microorganisms that constitute the microbiome are known to impact human physiology, both in health and in disease [21, 38, 20]. Thus, a central problem in CoDa is to learn the association $\mathbf{x}_i \mapsto y_i$, where $\mathbf{x}_i \in \Delta^{p-1}$ denotes the microbial composition, and $y_i$ denotes the disease status of the $i$th subject in a clinical study. For example, the composition of the gut microbiome, as recorded from a stool sample, can be predictive of colorectal cancer, which is the third most prevalent form of cancer [16, 33]. This observation offers the potential for a noninvasive alternative to traditional colonoscopy procedures used for early detection of colorectal cancer [56]. In turn, accurate predictive models for microbiome CoDa are a key stepping stone towards achieving this potential [39, 33]. Note that cancer is just one application; the demand for improved predictive models holds broadly more across medicine and biological science [33, 21]. For example, the microbiome has been linked to type 2 diabetes, Crohn's disease, obesity, and others [27, 62, 46, 20].

Classical techniques including logistic regression, support vector machines, and random forests have been widely used as predictive models for microbiome data [60, 6, 36]. More recently, specialized deep learning architectures such as DeepCoDa [48], MetaNN [37], and DeepMicro [41] have been developed. However, the capacity of these deep networks and other expressive models has been limited by the low sample size and high dimensionality of typical microbiome studies. These characteristics have also spurred the use of strong regularization through early stopping, weight decay and dropout, among others [57, 48, 37, 41]. However, no previous works have explored the use of data augmentation for CoDa,[2] which, as we shall demonstrate, provides a cheap and simple technique for boosting the performance of predictive models for this data modality.

Careful consideration of the sample space will motivate our novel data augmentation strategies for CoDa. Our work draws on foundational principles from the field of CoDa, such as the *Aitchison geometry* of the simplex [1], which we combine with popular techniques from the literature on data augmentation, such as Mixup [66] and CutMix [65]. This combination will lead us to define custom data augmentation strategies for CoDa, such as *Aitchison Mixup* and *Compositional CutMix*. In turn, incorporating these novel augmentations into existing supervised learning pipelines will result in consistent performance gains across a wide range of microbiome datasets, including a dozen standard benchmarks from the Microbiome Learning Repo [60]. These performance gains are particularly large for some deep models, for example, DeepCoDa gains more than 10% in test AUC for discriminating colorectal cancer from healthy controls, and over 20% for type 2 diabetes. The gains are also significant across other expressive model families, including random forests and gradient boosting machines. Overall, we set a new state-of-the-art on 8 out of 12 benchmark learning tasks, including clinically relevant disease prediction tasks. Of the remaining 4 tasks, 2 were easily separable, with 100% test accuracy irrespective of whether data augmentation was used. Importantly, our augmentations rarely hurt model performance, and in the few instances that this was the case, the drops were typically of only 1% in test AUC.

In addition to supervised learning, our novel augmentations will allow us to define *contrastive representation learning for CoDa*, which to the best of our knowledge, represents the first contrastive learning model for compositional data. Our novel data augmentations are at the core of this approach; the contrastive loss uses randomly augmented training examples to define a self-supervised optimization objective whose labels are generated from unlabelled data [26, 14, 9, 28]. In particular, our contrastive model is trained to discriminate between compositions that were generated as random augmentations of the same training example, and those that were generated as random augmentations of different training examples. Our implementation is adapted from SimCLR [9], but using a smaller network architecture together with our novel augmentations. Unlike SimCLR, we use our individual

---

[2]With the possible exception of dropout, which can be viewed as a form of data augmentation. Note that applications of dropout to CoDa, such as [37], use standard implementations that do not exploit the special structure of CoDa.

augmentation strategies in isolation, and find them sufficient to surpass previous representation learning approaches for microbiome CoDa [3, 41]. Altogether, we expect our data augmentations will enable significant future progress, possibly in combination with novel architectures, in both supervised and representation learning pipelines for microbiome CoDa.

## 2 Related Work

**Data augmentation:** numerous data augmentation strategies have been proposed in the context of image data [54] and text data [17]. LeNet-5 [35] used random shifts and resizing, and AlexNet [34] used reflections and color distortions, which have become widely adopted in computer vision [29]. More general purpose data augmentations include Mixup, which generates synthetic samples by taking convex combinations of training examples, and has been applied to both image and tabular data [66]. Mixup is related to an earlier method called SMOTE [8], which aims to address class imbalance by generating synthetic samples from the minority class, taken as convex combinations between nearest neighbors within the minority class. Random masking, whereby random input features are hidden during training, has also seen a diverse range of applications, including natural language processing (e.g., BERT [12]), computer vision (e.g., Cutout [13]), and tabular data (e.g., VIME [64]). In the context of CoDa, masking is related to *subcompositions*, which we will use to define an analogous augmentation on the simplex. CutMix [65] is akin to both masking and Mixup, in that random patches from different images are pasted together to generate new synthetic images. We note that a subsequent line of work has developed techniques to automatically select optimal data augmentations for a given dataset [10, 11], somewhat akin to hyperparameter optimization. While the data augmentation strategies that we present in the next Section could also be combined and finetuned in similar ways, such experimentation falls outside the scope of our work and is left to future research.

**Microbiome models:** random forests remain a strong baseline across many microbiome studies, due to their expressivity and robustness [57]. Gradient boosting machines such as XGBoost provide similarly strong performance and have also enjoyed significant adoption [57]. AutoML [30] has shown potential for microbiome data, with a recent approach called mAML achieving state-of-the-art results on several benchmarks [63]. mAML uses cross-validation to automatically select the best model class and hyperparameters for each learning task. In addition, several specialized deep learning architectures have been proposed for microbiome CoDa. DeepCoDa [48] introduced the log-bottleneck layer, which ensures the hidden units remain scale-invariant, a key desiderata in CoDa models. This architecture obtained strong predictive performance using weight decay regularization. MetaNN [37] employs a multilayer architecture regularized using dropout [55]. The authors of MetaNN also considered generating synthetic data from a negative binomial distribution, but found no performance improvements beyond the use of dropout. DeepMicro [41] uses a deep autoencoder architecture to learn low-dimensional representations of the microbial composition. In turn, these representations are fed to a downstream classifier trained with a supervised objective.

**Contrastive learning:** the goal of contrastive learning is to learn low-dimensional representations by optimizing some *pretext task*, where the objective function is similar to those used for supervised learning, but using labels that are derived from unlabelled data only [26]. Commonly, the pretext task is to discriminate augmented instances of the same training example from augmented instances of different training examples [14, 9, 28]. Recent contrastive learning architectures have enjoyed tremendous success, setting the state-of-the-art across a range of computer vision benchmarks [9, 28, 31, 7]. We highlight SimCLR [9] for its relative simplicity and strong empirical performance. This method uses a nonlinear projection head between the representations and the contrastive loss computed with normalized embeddings. Our own experiments on contrastive learning borrow these elements from SimCLR, but use a smaller network architecture together with our specialized data augmentations for CoDa.

## 3 Methods

In the following subsections, we introduce 3 novel data augmentation techniques for CoDa. Note that many variations of our augmentation schemes could be considered; the versions we present here are intended to be as concise as possible. Our goal is not to design augmentations that are empirically or theoretically optimal, but rather to demonstrate the effectiveness of our novel methodology by

establishing simple and performant baselines. For instance, whenever a random mixing parameter is required, we use a $U(0,1)$ distribution, even though other choices can likely result in increased performance. For clarity, we will focus on the classification setting; generalizing to regression problems is straightforward. We will use the notation $\mathbf{v}, \mathbf{x} \in \Delta^{p-1}$ for simplex-valued vectors, $\lambda \in \mathbb{R}$ a scalar, and $\mathcal{D} = \{\mathbf{x}_i, y_i\}$ our training data.

## 3.1 Aitchison Mixup

Aitchison [1] defined a Hilbert space structure on the simplex, known as the *Aitchison geometry*, with the following vector addition, scalar multiplication, and inner product:[3]

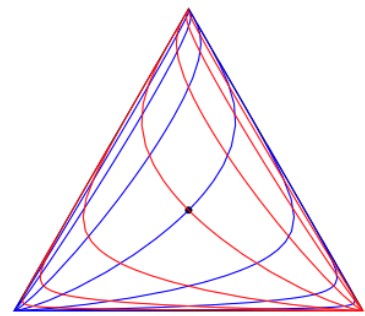

$$\mathbf{v} \oplus \mathbf{x} = \frac{1}{\sum_{j=1}^{p} v_j x_j} [v_1 x_1, \ldots, v_p x_p], \qquad (2)$$

$$\lambda \odot \mathbf{x} = \frac{1}{\sum_{j=1}^{p} x_j^{\lambda}} [x_1^{\lambda}, \ldots, x_p^{\lambda}], \qquad (3)$$

$$\langle \mathbf{v}, \mathbf{x} \rangle = \frac{1}{2p} \sum_{j=1}^{p} \sum_{k=1}^{p} \log\left(\frac{v_j}{v_k}\right) \log\left(\frac{x_j}{x_k}\right). \qquad (4)$$

This geometry provides a principled definition of linear combinations and a distance metric on the simplex; parallel and orthogonal lines are illustrated in Figure 1. The Aitchison geometry is closely related to the *centered-log-ratio* transformation [1], which is an inverse of the softmax and defines an isometry between $\Delta^{p-1}$ and Euclidean space [44]. Taken together, these properties form the basis of much of CoDa methodology [1, 44, 43, 18].

Figure 1: Orthogonal grid on $\Delta^2$, in the Aitchison sense [44]. The centroid of the simplex corresponds to the additive identity. The red lines are parallel and equally spaced by 1 unit in Aitchison distance, as are the blue lines. The red lines are also orthogonal to the blue lines.

Our first data augmentation strategy, which we call *Aitchison Mixup*, is based on taking linear combinations of the training points, in the Aitchison sense. While general linear combinations may be used, for simplicity we focus on intra-class pairwise convex combinations. Namely, each new datapoint is sampled as follows:

1. Draw a class $c$ from the class prior and draw $\lambda \sim U(0,1)$.
2. Draw two training points $i_1, i_2$ such that $y_{i_1} = y_{i_2} = c$, uniformly at random.
3. Set $\mathbf{x}^{\text{aug}} = (\lambda \odot \mathbf{x}_{i_1}) \oplus ((1 - \lambda) \odot \mathbf{x}_{i_2})$ and $y^{\text{aug}} = c$.

Put more succinctly, for each class we generate convex combinations of the points in that class, in the Aitchison sense. Note this augmentation strategy is a CoDa analogue of Mixup, which generates Euclidean convex combinations of images and tabular data [66]. Much like Mixup is capable of boosting predictive accuracy by regularizing models towards a linear decision boundary in the regions between training examples, Aitchison Mixup aims to enforce the equivalent regularization in the Aitchison geometry of the simplex.

Note that, once we restrict to convex combinations (as opposed to general linear combinations), our augmented data will remain inside the simplex regardless of whether we operate in the Euclidean or the Aitchison geometry. However, we found the latter to be more empirically effective, perhaps unsurprisingly given the associated vector space structure of the simplex.

We also note that the authors of Mixup [66] sampled $\lambda \sim Beta(\alpha, \alpha)$, where $\alpha$ is a hyperparameter, and took inter-class as well as intra-class convex combinations, setting $y^{\text{aug}} = \lambda y_{i_1} + (1 - \lambda) y_{i_2}$. While Aitchison Mixup can also incorporate such generalizations, we chose not to do so in our implementation because, as discussed above, our focus was on finding the simplest performant augmentation strategies for our new data modality.[4]

---

[3]Note that our method makes use of the Aitchison addition and scalar multiplication, but not the inner product, which is included for completeness.

[4]Inter-class combinations represent a significant complication to our methods because some classification libraries, including mAML, do not allow for non-binary outputs (without substantial rewrites) which are obtained when mixing labels from different classes. We also observed that, on DeepCoDa and MetaNN, using inter-class combinations did not improve performance.

## 3.2 Compositional Feature Dropout

In CoDa, a subcomposition of $\mathbf{x} \in \Delta^{p-1}$ refers to a lower-dimensional composition, $\mathbf{x}^{\text{sub}} \in \Delta^{k-1}$ with $k < p$, formed by taking a subset of the elements of $\mathbf{x}$ and renormalizing to a unit total. Namely, if $j_1, \ldots, j_k$ denotes a subset of the indices $\{1, \ldots, p\}$, the corresponding subcomposition is defined as:

$$\mathbf{x}^{\text{sub}} = \frac{1}{x_{j_1} + \cdots + x_{j_k}} [x_{j_1}, \ldots, x_{j_k}]. \tag{5}$$

We can generate augmented data by taking a *random* subcomposition of the training points; such a strategy is analogous to masking in language data or cropping in image data, or more generally, feature dropout [55]. Since our predictive models require inputs of fixed dimension $p$, rather than discarding elements of $\mathbf{x}$ we simply zero them out. Thus, each new datapoint is generated as follows:

1. Draw $\lambda \sim U(0, 1)$. Draw a training point $i$ uniformly at random and set $\tilde{\mathbf{x}} = \mathbf{x}_i$.
2. For each $j \in \{1, \ldots, p\}$, draw $I_j \overset{iid}{\sim} \text{Bernoulli}(\lambda)$, and set $\tilde{x}_j = 0$ if $I_j = 0$.
3. Set $\mathbf{x}^{\text{aug}} = \tilde{\mathbf{x}} / (\sum_{j=1}^p \tilde{x}_j)$ and $y^{\text{aug}} = y_i$.

In short, we zero out random entries of the training points and renormalize. Intuitively, this strategy encourages our predictive models to become robust to partially observed inputs. Note that, distinctively from feature dropout, our augmentation includes an additional renormalization to a unit total, and we therefore use the term *Compositional Feature Dropout*. Importantly, this renormalization ensures that the augmented samples remain in the support of the training data, i.e., the simplex.

Note that many CoDa models, including mAML and DeepCoDa, apply log transformations to their inputs, and therefore require that these be non-zero. For this reason, we implement a zero-replacement step where we add a small positive quantity to all the parts of each composition and renormalize. This small quantity is set to $1/L_i$, where $L_i$ corresponds to the library size from the high-throughput sequencing procedure. $L_i$ can simply be thought of as a large number such that $1/L_i$ is smaller than the non-zero components of $\mathbf{x}_i$. Note that this transformation is a standard preprocessing step in CoDa [50].

## 3.3 Compositional CutMix

Our third augmentation scheme, which we call *Compositional CutMix*, combines elements of our previous 2 augmentations. Like in Mixup, we generate new datapoints by combining pairs of training points from the same class. However, instead of combining these training points linearly (in the Aitchison sense), we take complementary subcompositions and renormalize. Namely, we generate each new datapoint as follows:

1. Draw a class $c$ from the class prior and draw $\lambda \sim U(0, 1)$.
2. Draw two training points $i_1, i_2$ such that $y_{i_1} = y_{i_2} = c$, uniformly at random.
3. For each $j \in \{1, \ldots, p\}$, draw $I_j \overset{iid}{\sim} \text{Bernoulli}(\lambda)$, and set $\tilde{x}_j = x_{i_1 j}$ if $I_j = 0$ or $\tilde{x}_j = x_{i_2 j}$ if $I_j = 1$.
4. Set $\mathbf{x}^{\text{aug}} = \tilde{\mathbf{x}} / (\sum_{j=1}^p \tilde{x}_j)$ and $y^{\text{aug}} = c$.

Note this strategy can be thought of as a CoDa analogue of *CutMix*, whereby new images are formed by pasting together patches from different training images [65]. Note also that in computer vision, CutMix and related methods take *local* image patches as opposed to randomly sampled pixels. Likewise, one could sample microbial subcompositions according to biologically relevant groupings, for example using a phylogenetic tree [50]. Such a strategy would require incorporating additional domain knowledge and is left to future work, but we expect it would further increase the quality of our data augmentations. This remark applies both to Compositional Feature Dropout and Compositional CutMix. With regard to inter-class combinations, a similar discussion as in Section 3.1 applies.

## 4 Experiments

### 4.1 Supervised learning

We evaluate our augmentation strategies on 12 standard binary classification tasks taken from the Microbiome Learning Repo [60]. These datasets comprise various disease and phenotype prediction

Table 1: Evaluation benchmark consisting of 12 binary classification tasks taken from the Microbiome Learning Repo [60], after filtering to datasets containing at least 100 samples with at least 50 in each class. For each task we show the number of samples ($n$), the number of features ($p$), a description of the two classes and the number of samples in each, together with a reference to the original studies that each dataset was obtained from.

| Task | $n$ | $p$ | Class 1 / Class 2 | # in 1 | # in 2 | Reference |
|------|------|------|-------------------|--------|--------|-----------|
| 1 | 140 | 992 | Crohn's disease / Without (ileum) | 78 | 62 | [20] |
| 2 | 160 | 992 | Crohn's disease / Without (rectum) | 68 | 92 | [20] |
| 3 | 2070 | 3090 | Gastrointestinal tract / Oral | 227 | 1843 | [38] |
| 4 | 180 | 3090 | Female / Male | 82 | 98 | [38] |
| 5 | 404 | 3090 | Stool / Tongue (dorsum) | 204 | 200 | [38] |
| 6 | 408 | 3090 | Subgingival / Supragingival plaque | 203 | 205 | [38] |
| 7 | 172 | 980 | Healthy / Colorectal cancer | 86 | 86 | [33] |
| 8 | 124 | 2526 | Diabetes / Without | 65 | 59 | [46] |
| 9 | 130 | 2579 | Cirrhosis / Without | 68 | 62 | [47] |
| 10 | 199 | 660 | Black / Hispanic | 104 | 95 | [52] |
| 11 | 342 | 660 | Nugent score high / Low | 97 | 245 | [52] |
| 12 | 200 | 660 | Black / White | 104 | 96 | [52] |

tasks, including colorectal cancer [33], type 2 diabetes [46], Crohn's disease [20], and cirrhosis [47], as well as multiple body sites including the gut, skin, oral cavity, airways, and vagina [38, 52]. As such, this benchmark provides a comprehensive evaluation for predictive models of the human microbiome [63, 48, 24]. More details on these 12 learning tasks can be found in Table 1; note this benchmark is constructed from the original Microbiome Learning Repo by filtering datasets that contain a minimum sample size of 100, with at least 50 in either class.

For each learning task, we take 20 independent 80/20 train/test splits and we fit Random Forest, XGBoost, mAML [63], DeepCoDa [48], and MetaNN [37], first to the original training data, then on 3 augmented training sets obtained using our 3 augmentation strategies. Thus, we train a total of $12 \times 20 \times 5 \times 4 = 4\,800$ models.[5] We evaluate test performance using ROC AUC, and we note that our datasets do not present severe class imbalance (with the exception of tasks 3 and 11, both of which were well separated by all our classifiers and therefore had no effect on the overall comparison).

For the augmented training sets, we generated 10 times as many synthetic samples as there were original training examples. The factor of 10 was chosen so as to obtain a relatively large augmented sample, in order to reduce the sampling variance from our random augmentations. In turn, we compensate for the fact that our augmented data is then much more numerous than our original training data, by downweighting the synthetic samples by a factor of 10; the total weight of the original and synthetic data is then equal to $1/2$ each.

Our results are shown in Table 2 (Aitchison Mixup) in the main text, as well as Tables 5 (Compositional Feature Dropout), and 6 (Compositional CutMix) in the Appendix:

- Table 2 shows that Aitchison Mixup improved the test performance of existing models across a large majority of learning tasks. On average, we obtained a 5% gain in test AUC for DeepCoDa, 2% for random forests and XGBoost, and 1% for mAML. MetaNN remained flat, but this was also the worst performing model overall. Importantly, in the few instances where Aitchison Mixup hurt the performance of a model, the loss was typically of just 1%.
- Table 5 shows that Compositional Feature Dropout also result in consistent performance improvements across most models and tasks, though to a slightly lesser degree than Aitchison Mixup. On average, DeepCoDa enjoyed a 3% gain in test AUC, random forests 2%, and XGBoost and MetaNN both 1% (mAML remained flat).
- Table 6 shows that Compositional CutMix obtained the strongest classification performance out of our 3 augmentation strategies on microbiome CoDa. On average, DeepCoDa saw a 5% gain in test AUC, with 2% for random forests and XGBoost, and 1% for mAML and MetaNN. Moreover, models trained with Compositional CutMix set a new state-of-the-art on 8 out of 12 tasks, including colorectal cancer (0.76), type 2 diabetes (0.72), and Crohn's

---

[5]We train these models in parallel on a CPU cluster.

Table 2: Data augmentation performance for Aitchison Mixup. We show the test AUC, averaged over 20 train/test bootstraps, for each learning task and predictive model, trained with and without data augmentation. Bold numbers indicate whether the version with or without augmentation performed best. Underlined numbers indicate the overall best model for that task. Models trained with Aitchison Mixup consistently outperformed those without (with the possible exception of MetaNN, which performed worst across the board). Error bars are given in Appendix A.

| Task | RF | Aug | XGB | Aug | mAML | Aug | DeepCoDa | Aug | MetaNN | Aug |
|------|------|------|------|------|------|------|----------|------|--------|------|
| 1 | 0.72 | **_0.79_** | 0.76 | **0.79** | 0.72 | **0.74** | 0.73 | **_0.79_** | **0.74** | **0.74** |
| 2 | 0.78 | **0.82** | **0.81** | 0.80 | **0.80** | **0.80** | 0.78 | **_0.83_** | **0.74** | **0.74** |
| 3 | **1.00** | **1.00** | **1.00** | **1.00** | **1.00** | **1.00** | **1.00** | **1.00** | **1.00** | **1.00** |
| 4 | 0.60 | **_0.64_** | 0.57 | 0.57 | 0.56 | **0.58** | **0.58** | 0.58 | 0.50 | **0.51** |
| 5 | **1.00** | **1.00** | **1.00** | **1.00** | **1.00** | **1.00** | **1.00** | **1.00** | **1.00** | **1.00** |
| 6 | 0.81 | **0.83** | 0.82 | **0.83** | **_0.84_** | 0.83 | 0.78 | **0.82** | 0.75 | **0.76** |
| 7 | **0.68** | 0.67 | 0.67 | **0.69** | 0.73 | **_0.74_** | 0.63 | **0.73** | **0.59** | 0.54 |
| 8 | 0.62 | **0.65** | 0.66 | **0.68** | 0.64 | **0.65** | 0.45 | **_0.70_** | **0.64** | **0.64** |
| 9 | **0.93** | **0.93** | 0.94 | **_0.95_** | 0.92 | **0.93** | 0.84 | **0.90** | 0.76 | **0.82** |
| 10 | 0.53 | **0.60** | 0.57 | **0.61** | 0.61 | **0.62** | 0.62 | **_0.63_** | **0.63** | 0.61 |
| 11 | **0.98** | **0.98** | **0.98** | **0.98** | **0.98** | **0.98** | **0.98** | **0.98** | **0.96** | 0.95 |
| 12 | 0.55 | **0.61** | 0.58 | **0.65** | **0.61** | **0.61** | **_0.66_** | 0.65 | 0.58 | **0.60** |
| Mean | 0.77 | **0.79** | 0.78 | **0.80** | 0.78 | **0.79** | 0.75 | **0.80** | **0.74** | **0.74** |

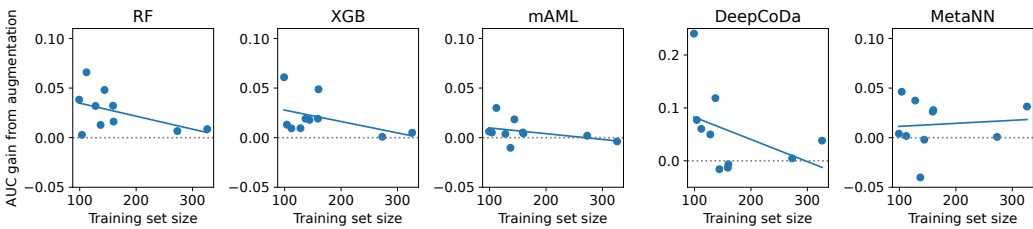

Figure 2: Difference in test AUC between models trained with Compositional CutMix, and those without, shown as a function of training set size. Each point represents one of our 12 benchmark learning tasks (note that tasks 3 and 5 are removed from the plot, since they already enjoyed 100% classification accuracy prior to applying any data augmentation). The linear trendlines show a somewhat greater outperformance on the smaller datasets relative to the larger datasets.

disease (0.78 and 0.83). Of the remaining 4 tasks, 3 were tied and only on 1 was the best model one that did not use Compositional CutMix.

Figure 2 shows the effect of data augmentation on test performance as a function of the number of samples in the training set, for Compositional CutMix. We note that datasets with smaller sample sizes tended to benefit more from data augmentation. This suggests that our methodology may prove beneficial to many microbiome research projects, where datasets with low-hundreds of samples are commonplace. Figure 3 shows a similar scatterplot, as a function of the dataset dimensionality rather than sample size; the gains from augmentation appear consistent across lower- and higher-dimensional microbiome CoDa (we observe similar trends Aitchison Mixup and Compositional Feature Dropout, and we omit the corresponding scatterplots).

In addition, we evaluated the impact of our data augmentations on the expected calibration error (ECE) [25] of our models. Previous works have noted that increasingly flexible predictive models such as deep neural networks, while enabling greater predictive accuracy, tend to become overconfident in their predictions, degrading uncertainty quantification [25]. In Appendix A, we show the ECE obtained by our models, and we verify that our data augmentations do not hurt model calibration overall (in fact, some modest improvements are obtained).

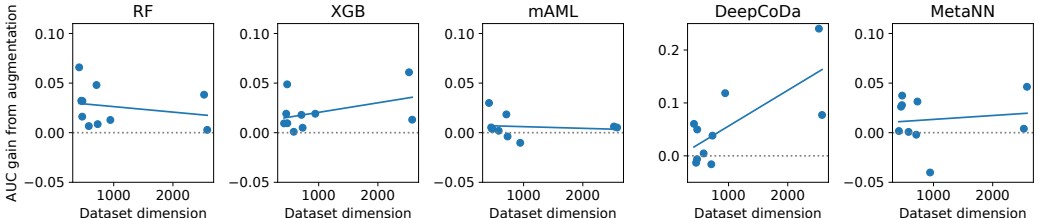

Figure 3: Similar to Figure 2. Difference in test AUC between models trained with Compositional CutMix, and those without, shown as a function of dataset dimension.

## 4.2 Contrastive representation learning

We have shown that our novel data augmentation strategies provide consistent performance gains across supervised training pipelines. Next, we use our data augmentations in order to apply contrastive representation learning to compositional data for the first time (to the best of our knowledge). Note that our experiment provides a simple proof of concept showing that contrastive learning is capable of learning useful representations of microbiome CoDa, which result in better downstream classification performance than previous deep representation learning approaches. However, more extensive experiments will be required to validate the performance of contrastive learning across a wider range of model architectures for CoDa.

As a point of comparison we take DeepMicro [41], a deep autoencoder designed specifically for microbiome data. DeepMicro includes an encoder and a decoder that are trained jointly by minimizing reconstruction error. The encoder outputs are used as a low-dimensional representation that is passed to a downstream classifier. Our contrastive model discards the decoder and trains the encoder using the contrastive loss from SimCLR [9]. This loss function requires pairs of *positive* and *negative* datapoints, where positive pairs are constructed by taking synthetic augmentations of the same training example and negative pairs are constructed by taking synthetic augmentations of different training examples.[6] In the context of microbiome CoDa, it is our novel augmentations that allow us to define positive and negative pairs required for contrastive learning. Full detail on model architecture and implementation is provided in Appendix B.

After applying unsupervised pretraining (reconstructive or contrastive), we evaluate the quality of learned representations under two standard protocols: linear evaluation and finetuning. In linear evaluation, we freeze the encoder weights and train a linear classification head using the supervised cross-entropy loss on the original training set. In finetuning, the linear classification head is trained jointly with the encoder network, again using the supervised cross-entropy loss. The finetuning protocol is evaluated both with and without using additional data augmentation (i.e., passing additional synthetic examples to the supervised loss). These evaluation protocols are applied for both DeepMicro and our contrastive model, in addition to a "no pretraining" control. This control is a randomly initialized encoder network and linear head with the same architecture, trained only on the supervised objective (i.e., a neural network that did not enjoy the benefit of unsupervised pretraining).

The results are shown in Table 3; the representations learned by our contrastive model using Compositional Feature Dropout consistently outperform those learned by DeepMicro, both in linear evaluation and finetuning. Note that finetuning with augmentation performs slightly better than without, suggesting that the benefits of supervised training with our data augmentations can be combined with the benefits of augmentation as part of contrastive learning. Importantly, note that the comparison to DeepMicro is conservative, in the sense that we replicated the architecture and simply changed the pretraining objective from reconstruction error to contrastive loss; tuning the encoder architecture itself under a contrastive objective would likely lead to further performance improvements. We conclude that Compositional Feature Dropout provides a valuable data augmentation for contrastive learning on the simplex.

Note that our other 2 augmentation strategies, Aitchison Mixup and Compositional CutMix, may also be used for contrastive learning. However, these augmentations require paired training examples to generate each synthetic sample, and the implementation is therefore slightly more involved. In

---

[6]Note such pairs are defined without reference to the (possibly unknown) true labels.

Table 3: Representation learning performance for DeepMicro and contrastive learning with Compositional Feature Dropout. For each task, we show the test AUC, averaged over 20 train/test bootstraps, under the linear evaluation protocol and finetuning. We also show the effect of using additional data augmentation as part of supervised training during the finetuning stage ("Finetuning with Aug"). In addition, a randomly initialized encoder network of the same architecture with no pretraining is shown for comparison. Note that the representations obtained via contrastive learning consistently achieve higher AUC than those learned by DeepMicro. Error bars are given in Appendix A. No Pre=no pretraining, DM=DeepMicro autoencoder pretraining, Contr=contrastive pretraining (ours).

| Task | Linear Evaluation | | | Finetuning | | | Finetuning with Aug | | |
| | No Pre. | DM | Contr. | No Pre. | DM | Contr. | No Pre. | DM | Contr. |
|------|---------|------|--------|---------|------|--------|---------|------|--------|
| 1 | 0.59 | 0.68 | **0.76** | 0.72 | 0.75 | **0.77** | **0.77** | 0.76 | **0.77** |
| 2 | 0.67 | 0.76 | **0.80** | 0.76 | 0.76 | **0.79** | 0.78 | 0.79 | **0.80** |
| 3 | **1.00** | **1.00** | **1.00** | **1.00** | **1.00** | **1.00** | **1.00** | **1.00** | **1.00** |
| 4 | 0.47 | 0.53 | **0.59** | 0.50 | 0.53 | **0.55** | 0.52 | **0.56** | **0.56** |
| 5 | **1.00** | **1.00** | **1.00** | **1.00** | **1.00** | **1.00** | **1.00** | **1.00** | **1.00** |
| 6 | 0.75 | 0.76 | **0.77** | **0.77** | 0.76 | **0.77** | 0.78 | 0.78 | 0.78 |
| 7 | 0.59 | **0.68** | 0.63 | **0.59** | **0.59** | 0.57 | 0.57 | **0.62** | 0.60 |
| 8 | 0.65 | **0.66** | **0.66** | 0.66 | **0.68** | **0.68** | 0.66 | 0.66 | **0.68** |
| 9 | 0.72 | 0.71 | **0.86** | 0.76 | 0.77 | **0.82** | 0.79 | 0.80 | **0.82** |
| 10 | 0.53 | 0.58 | **0.62** | 0.60 | 0.61 | **0.63** | 0.62 | 0.60 | **0.63** |
| 11 | 0.96 | **0.98** | **0.98** | **0.98** | **0.98** | **0.98** | **0.97** | **0.97** | **0.97** |
| 12 | 0.66 | **0.68** | 0.66 | 0.62 | **0.64** | **0.64** | 0.62 | **0.65** | **0.65** |
| Mean | 0.72 | 0.75 | **0.78** | 0.74 | 0.76 | **0.77** | 0.76 | 0.77 | **0.78** |

particular, we generate each pair of positive samples by taking one pair of training examples, and drawing two random combinations (in the Aitchison Mixup or the Compositional CutMix sense) of that one pair. The training examples are randomly partitioned into pairs at each epoch, and negative samples correspond to those that originate from separate pairs. The results are provided in Appendix A and show similarly strong performance to Table 3.

## 4.3 Additional benchmarks

Further to the 12 microbiome classification tasks from the previous sections, we analyze 8 additional compositional datasets, which were selected to represent a diverse set of applications that are commonly encountered by CoDa practitioners. Note that, while these datasets correspond to a wide range of scientific domains, they have all been analyzed as case studies specifically in the CoDa literature. The datasets can be summarized as follows:

- *Glass*: a dataset from the UCI ML repo, on the elemental composition of glass particles of different types of origin. Such particles are routinely recovered from the clothing of criminal suspects; the goal is to provide forensic evidence by classifying the type of origin (broken window vs. other) given the composition recorded in a multielement analysis [15, 58].
- *Bayesite*: a classical materials science dataset due to Aitchison [1]. The goal is to predict the permeability of fibreboard from the mix of its ingredients.
- *Serum*: another classical dataset due to Aitchison [1], where the composition of blood serum protein is used to discriminate between two known diseases.
- *Hydrochem*: a hydrochemical dataset measuring the composition of water samples across sites in the Llobregat river basin [42]. Classification targets are formed by separating upstream and downstream sites.
- *Jura*: a geochemical dataset from the Swiss Jura, where rock types (Kimmeridgian vs. Sequanian) are classified from the chemical composition of soil samples [2].
- *Metabolites*: gut metabolomic profiles for subjects with and without inflammatory bowel disease [19, 49]. The goal is to diagnose disease from the metabolite composition.
- *MicroRNA*: contains microRNA expression data for different types of primary breast cancer [61, 49]. The goal is to discriminate between breast cancer subtypes (Luminal A vs. Luminal B) from the molecular profile of each tumor.

Table 4: Data augmentation performance for Aitchison Mixup on non-microbiome CoDa benchmarks. We show the test AUC, averaged over 20 train/test bootstraps, for each non-microbiome dataset and predictive model, trained with and without data augmentation. Bold numbers indicate whether the version with or without augmentation performed best. Underlined numbers indicate the overall best model for that task. Models trained with Aitchison Mixup consistently outperformed those without.

| Dataset | $n/p$ | RF | Aug | XGB | Aug | DeepCoDa | Aug | NN | Aug |
|---|---|---|---|---|---|---|---|---|---|
| Glass [15] | 213/8 | 0.84 | **0.86** | 0.84 | **0.86** | 0.83 | **0.84** | 0.84 | 0.84 |
| Bayesite [1] | 20/5 | **0.86** | **0.86** | 0.84 | **0.90** | **0.69** | 0.54 | 0.53 | **0.58** |
| Serum [1] | 30/4 | 0.81 | **0.83** | 0.80 | **0.82** | 0.62 | **0.82** | **0.87** | 0.82 |
| Hydrochem [42] | 246/14 | **1.00** | **1.00** | **1.00** | **1.00** | 0.96 | **0.99** | 0.95 | 0.95 |
| Jura [2] | 147/7 | 0.93 | **0.94** | 0.92 | **0.94** | 0.83 | **0.93** | 0.75 | 0.72 |
| Metabolites [19] | 220/885 | 0.95 | **0.96** | 0.94 | 0.94 | 0.84 | **0.94** | 0.92 | **0.93** |
| MicroRNA [61] | 717/188 | **0.90** | **0.90** | 0.91 | **0.92** | 0.87 | **0.90** | 0.90 | 0.90 |
| Coffee [32] | 30/15 | 0.92 | **0.93** | 0.80 | **0.83** | 0.71 | **0.77** | 0.65 | **0.69** |
| Mean | - | 0.90 | **0.91** | 0.88 | **0.90** | 0.79 | **0.84** | 0.80 | **0.81** |

- *Coffee*: a nutrition dataset describing the chemical composition of commercially available coffee samples [32]. Classification targets are taken as different blends of coffee.

In modeling these datasets, we apply the *centered logratio* transformation to the model inputs [1] and follow the experimental protocol described in Section 4.1, except we no longer test mAML and MetaNN (which were not intended for non-microbiome data), and we replace the latter with a small neural network with one hidden layer of 32 units and ReLU activations, denoted as NN (note we choose to keep this architecture very simple as it is trained on several heterogeneous dataset).

The results are shown in Table 4 for Aitchison Mixup. As was the case for our microbiome benchmark, we again see that our data augmentation results in consistent performance gains across a wide range of classifiers. Indeed, on most datasets (6 out of 8) the best performing model was one that used Aitchison Mixup, and only on a single dataset (Serum) was the best model (NN) one trained without augmentation. Note that, in contrast to our microbiome benchmarks, Table 4 shows a strong outperformance for tree-based models. Importantly, Aitchison Mixup never hurt the performance of any of our tree-based models, and only did so rarely for our network models. We observe similarly positive results (albeit slightly weaker) for Compositional Feature Dropout and Compositional CutMix, provided in Tables 7 and 8 from the Appendix, respectively.

## 5 Conclusion

By combining ideas from data augmentation with the principles of CoDa, we have defined 3 novel augmentation strategies for CoDa: Aitchison Mixup, Compositional Feature Dropout, and Compositional CutMix. Our augmentations offer cheap performance gains across a wide range of benchmark datasets, advancing the state-of-the-art on standard classification benchmarks for colorectal cancer, type 2 diabetes, and Crohn's disease. While we find all 3 strategies performant across a wide range of benchmark datasets, Aitchison Mixup and Compositional CutMix performed best on microbiome CoDa (with a slight outperformance for the latter), and Aitchison Mixup performed best on non-microbiome CoDa. In addition, our data augmentations allowed us to define the first contrastive learning model for CoDa, which we show improves on existing representation learning frameworks for microbiome data. Data augmentation and contrastive learning have enjoyed tremendous success in other application domains such as computer vision; our novel methodology aims to help spur similar developments in the fields of CoDa and the microbiome.

## Acknowledgments and Disclosure of Funding

We thank Samuel Lippl, Richard Zemel, and the anonymous reviewers for helpful discussions, and the Simons Foundation 542963, Sloan Foundation, McKnight Endowment Fund, NSF DBI-1707398, and the Gatsby Charitable Foundation for support.

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
