# OpenReview forum: "Data Augmentation for Compositional Data: Advancing Predictive Models of the Microbiome"
_NeurIPS.cc/2022/Conference — NeurIPS 2022 Accept_

### Official Review · Reviewer_7STz · 2022-07-11

**Rating:** 7
**Confidence:** 4
**Soundness:** 3 good
**Presentation:** 3 good
**Contribution:** 3 good

**Summary:**

This paper looks at how one can do data augmentation on compositional data, in order to improve generalization of supervised learning on data of this type, as well as enable contrastive self-supervised representation learning.  The method utilizes vector space structures on the simplex provided by Aitchison geometry to define an augmentation scheme that can compose multiple compositional data types together, similar to what is done with the MixUp method for data types that can be combined together linearly.  They explore augmenting human microbiome data, which is characterized as the fraction of various microbe species in a person’s gut.  The paper finds improvement (some very convincing, others marginal) when classifying various phenotypes based on their microbiome using their data augmentations.  They also study contrastive methods, similar to SimCLR, which are enabled by the compositional data augmentations.

**Questions:**

1. How does the method perform on other compositional data types from other fields?  Why did you focus so much on the microbiome?
2. In the Aitchison Mixup section, step 2: what is the reasoning for requiring the two datapoints you combine to have the same class?  If you follow MixUp, they advocate doing a similar linear combination of labels.
3. (Minor) Did you try other simple augmentations on compositional data, such as shuffling?
4. (Minor) I might have missed it, but did you actually use the Aitchison inner product (Eq 4)?


**Limitations:**

The authors didn’t explicitly address limitations or societal impact, although they did describe some improvements they might expect to make on the type of data augmentations, e.g., using domain specific knowledge.  I can’t really think of a major negative societal impact of this paper, though.

**Strengths And Weaknesses:**

Strengths:

1. I like how the method utilizes the pre-existing Aitchison geometry to keep combinations of data points on the simplex.
2. It is good to see quantification via retraining on multiple train/test splits, compared to variance seen by just modifying the random seed.
3. I have not seen (nor could I find when reviewing the literature) anyone doing SimCLR-style contrastive learning on compositional data; the authors claim this is the first instance they could find, too.


Weaknesses:

1. I don’t really understand why the paper has focused so much on the human microbiome.  As the authors point out, compositional data is ubiquitous & their data augmentation methodology does not use any domain-specific knowledge or anything else that requires the focus to be on the microbiome.  I find the narrow positioning of an otherwise general method is doing the paper a major disservice.

---

> ### Author Response · Authors · 2022-08-02
> **Additional benchmarks on non-microbiome CoDa have been added and other comments have been addressed**
>
> Many thanks for the kind and constructive feedback.
>
> >Weaknesses:
> >1.	I don’t really understand why the paper has focused so much on the human microbiome. As the authors point out, compositional data is ubiquitous & their data augmentation methodology does not use any domain-specific knowledge or anything else that requires the focus to be on the microbiome. I find the narrow positioning of an otherwise general method is doing the paper a major disservice.
> >
> >Questions:
> >1.	How does the method perform on other compositional data types from other fields? Why did you focus so much on the microbiome?
>
>
> We agree that our work would benefit from expanding our experiments to other common types of compositional datasets, a point that was also raised in other reviews.
> That said, we also note that the human microbiome remains one of the biggest and most high-impact areas of application for CoDa methodology.
>
>
> Thus, in addition to our 12 microbiome classification tasks, we have now added 8 separate non-microbiome compositional datasets to our benchmarks. These datasets are well-studied in the CoDa literature and were selected to represent a diverse mix of scientific applications that are relevant to CoDa practitioners, including materials science, hydrochemistry, geochemistry, metabolomics, oncology, and nutritional science.
>
> The full detail regarding these datasets and corresponding experimental results is provided in a new Subsection (see 4.3 in the updated manuscript).
> We highlight the following key results:
> - As was the case on our microbiome benchmark, we find that our data augmentations again deliver consistent improvements in classification performance across a large majority of our non-microbiome datasets and predictive models.
> - On 6 out of 8 non-microbiome datasets, the best performing model was one using Aitchison Mixup.
> - On these non-microbiome datasets, tree-based models strongly outperformed network models. Aitchison Mixup never hurt the performance of these tree-based models (and did so rarely for our network models).
>
>
> >2	In the Aitchison Mixup section, step 2: what is the reasoning for requiring the two datapoints you combine to have the same class? If you follow MixUp, they advocate doing a similar linear combination of labels.
>
>
> Thank you for flagging; we have added an explanatory note towards the end of Section 3.1 of the manuscript.
> The main reason for restricting to intra-class combinations was ease of implementation; some classification libraries (including mAML) do not allow for non-binary outputs (without major rewrites) which are obtained when mixing labels from different classes, and as stated at the top of Section 3 we tried hard to keep our methods simple and user-friendly (as well as principled and performant).
> That said, we welcome the use of inter-class combinations in future applications (provided the implementation cost is reasonable), but in the cases that we tried them out (DeepCoDa and MetaNN) they did not improve performance.
>
>
>
> >3	(Minor) Did you try other simple augmentations on compositional data, such as shuffling?
>
>
> Other augmentations that we tried include "Multinomial Resampling", which can be thought of as a CoDa analogue of image blurring, and is presented in Appendix C.
> We did not explore shuffling specifically, because it seems unclear to us how such a transformation might be interpreted in the context of compositional data.
>
>
> >4	(Minor) I might have missed it, but did you actually use the Aitchison inner product (Eq 4)?
>
> Our method only uses the Aitchison vector addition and scalar multiplication, not the inner product. Eq. 4 is simply there for completeness and to highlight the Hilbert space structure. We have added a clarifying footnote to the manuscript.

---

### Official Review · Reviewer_ymr8 · 2022-07-11

**Rating:** 7
**Confidence:** 4
**Soundness:** 3 good
**Presentation:** 4 excellent
**Contribution:** 3 good

**Summary:**

The paper presents three data augmentation methods for compositional data. The first method (Aitchison Mixup) takes convex combinations of samples *within* the same class. The second method (Random Subcompositions - similar to applying dropout on features) takes an input, samples a subset of features, makes the values zero, and renormalizes the new point to be on the simplex. The third method (Compositional CutMix) samples two points from the same class, performs CutMix between them, and renormalizes the new point to be on the simplex.

All three methods are adaptations of existing methods to Compositional Data (i.e., use the Simplex constraint) and are applied only intra-class (while the original Mixup and CutMix are also applied inter-class).

The paper's contribution is to adapt existing data augmentation methods to Compositional Data and show consistent performance improvement for classification tasks.

**Questions:**

I provide context for all questions in the “weaknesses” section

1. When to use each of the proposed methods?
2. Is any trend by comparing Figures 2 and 3 computed for all three proposed methods?
3. How does contrastive pre-training compare with training from scratch using data augmentation, and contrastive pre-training + data augmentation fine-tuning (my comment about incomplete Section 4.2)?
4. What's the motivation for applying the data augmentation only *intra-class* and not *inter-class* like the original Mixup and CutMix? Consider mentioning this in the paper as well. Also, highlight in the paper that the proposed methods are applied only *intra-class*.
5. Why does the performance after fine-tuning the model pre-trained using contrastive learning decrease from 0.78 to 0.77 (Table 5), while it increases for the other two models?
6. What's the masking percentage (for Random Subcompositions) in the contrastive learning experiments?


**Limitations:**

I couldn't find where the authors mentioned the limitations of the current work. One limitation I identified is proposing three (similar) methods without offering some rule-of-thumb when to use each technique.

I don't see any potential negative societal impact.

**Strengths And Weaknesses:**

# Strengths
- **Well-written paper**. The paper is easy to follow, and the authors provide good details about the task of learning from compositional data and the proposed methods.
- **Very simple methods, with good empirical results**. The proposed methods are relatively trivial to implement and don't add any computational overload.


# Weaknesses
- **The contrastive learning experiments seem incomplete**. Although interesting, the experiment from Section 4.2 is evaluated too simplistically and leaves unanswered questions. Table 5 compares a standard MLP encoder that is 1) randomly initialized, 2) pre-trained using an encoder-decoder and 3) pre-trained with contrastive learning and the proposed data augmentation. Table 5 shows that pre-training using contrastive learning provide more linearly separable representations than the representations learned by an autoencoder.
	- However, Section 4.2 raises the question: "When should use contrastive pre-training rather than standard supervised training with the proposed data augmentation (like in section 4.1)?" because the results from Table 5 aren't compared with the performance of an encoder trained using data augmentation (like in section 4.1).
	- I suggest adding in Table 5 the following numbers:
		1. encoder from DeepMicro + classification head-trained from scratch using data augmentation (similar to Table 5)
		2. encoder from DeepMicro pre-trained using contrastive learning + fine-tuned (with a classification head) using data augmentation — this answer the question: "Is contrastive pre-training complementary to training using data augmentation?"
- **Missing guideline for using the proposed methods on new datasets**. All three proposed data augmentation methods seem to provide a similar average performance improvement. It would be helpful to offer rule-of-thumb guidelines for choosing between the three methods (without trying all three methods). Please answer the question: "When should one use one method versus the other two?". Did you notice any pattern of a method working better for specific tasks/datasets?
	- **Possible solution**: Reproduce Figures 2 and 3 (performance improvement as a function of dataset size and the number of features) for all three proposed methods. Possibly, a pattern will emerge. If the results are different, add all figures to the appendix. If the results are similar, mention this.
	- **(Small, but related) Tables 2/3/4 are too similar** and the paper doesn't mention any trend difference between them. Although lines 237-251 describe the results, there's no explicit mention of the differences in the results for the three tables. Unless you specify clear trend differences, two tables can be moved to the appendix, and only the mean performance can be kept in the main paper.
- **(Small) Missing relation to prior work**: The proposed "Aitchison Mixup" is equivalent to performing SMOTE [1] in the Aitchison sense and shall be called Aitchison SMOTE. When Mixup takes convex combinations within the *same* class, it's equivalent to SMOTE. Please check the SMOTE method and the second paragraph of the related work on the mixup paper. If you use SMOTE, then please add it to the related work.

# Minor changes
- Line 30: Add examples of compositions data beyond naming domains (e.g., geochemistry -> proportion of chemical in a geological sample; economics -> proportion of GDP attributed to each industry), to make it easier for readers new to the area.
- Where are the limitations of the current work written? I couldn't find them in Section 3 (as mentioned in the checklist)
- Reference the peer-reviews published version of the papers (e.g., Mixup appeared in ICLR, and you've referenced the arXiv article)
- Line 250: provide the number of the SOTA tasks in parenthesis for easy reading
- Figure 3 from the appendix is really nice. Consider adding it to the main paper.
- The method "Random Subcompositions" could be renamed to "Compositional Feature Dropout", because it performs dropout on features, then renormalizes the feature to present the property of compositional data.

---

> ### Author Response · Authors · 2022-08-02
> **More complete contrastive experiments have been added and other points have been addressed**
>
> Many thanks for the detailed and constructive feedback.
>
> >• The contrastive learning experiments seem incomplete. [...] 3.	How does contrastive pre-training compare with training from scratch using data augmentation, and contrastive pre-training + data augmentation fine-tuning (my comment about incomplete Section 4.2)?
>
> We have added these comparisons to Table 5, as suggested (note it is now Table 3, after moving Tables 3 & 4 to the Appendix per your other suggestion).
> We find that the addition of data augmentation during finetuning typically improves performance, indicating that the benefit of data augmentation for supervised training may be somewhat additive to that obtained from contrastive pretraining (see discussion in Section 4.2).
> This is a promising result, but we think more extensive work focused specifically on contrastive learning will be required in order to validate this learning paradigm across a broader range of model architectures for CoDa.
> We leave this direction to future work, noting that our contrastive experiment provides a simple proof-of-concept, highlighting a cool application of our novel data augmentations.
>
> >•	Missing guideline for using the proposed methods on new datasets. [...] 1.	When to use each of the proposed methods?
>
> We have added such guidelines in Section 5 (Conclusion).
> While we found all 3 augmentation strategies performant across a wide range of benchmark datasets, Aitchison Mixup and Compositional CutMix performed best on microbiome CoDa (with a slight outperformance for the latter), and Aitchison Mixup performed best on non-microbiome CoDa. On the other hand, Compositional Feature Dropout has implementation advantages for contrastive learning.
> The trends observed in Figures 2 and 3 persist across data augmentations, as has been noted in Section 4.1.
>
>
> >4	What's the motivation for applying the data augmentation only intra-class and not inter-class like the original Mixup and CutMix? Consider mentioning this in the paper as well. [...]
>
> Thank you for flagging; we have added an explanatory note towards the end of Section 3.1 (and 3.3) of the manuscript.
> The main reason for restricting to intra-class combinations was ease of implementation; some classification libraries (including mAML) do not allow for non-binary outputs (without major rewrites) which are obtained when mixing labels from different classes, and as stated at the top of Section 3 we tried hard to keep our methods simple and user-friendly (as well as principled and performant).
> That said, we welcome the use of inter-class combinations in future applications (provided the implementation cost is reasonable), but in the cases that we tried them out (DeepCoDa and MetaNN) they did not improve performance.
>
> >•	(Small) Missing relation to prior work: The proposed "Aitchison Mixup" is equivalent to performing SMOTE [1] in the Aitchison sense [...]
>
> Thank you, we have added the reference. Note, however, that SMOTE was proposed as a method for addressing class imbalance, combining points from the minority class only, unlike Mixup. Another key difference is that SMOTE takes combinations only among K nearest neighbors, as opposed to sampling across the training set. On balance, "SMOTE Mixup" does not seem more faithful to our method than "Aitchison Mixup", but we are open to other suggestions if you have them (note that, as of the update, we explicitly state that inter-class combinations may be used even though our implementation did not do so for the reasons discussed).
>
> >5	Why does the performance after fine-tuning the model pre-trained using contrastive learning decrease from 0.78 to 0.77 (Table 5), while it increases for the other two models?
>
> This is most likely due to overfitting, which is perhaps unsurprising given the capacity of the DeepMicro encoder network relative to the dataset sizes in our microbiome benchmarks. Note that other instances of finetuning underperforming linear evaluation in self-supervised learning have been reported (e.g., Kumar et al., ICLR 2022). In our case, the result suggests that contrastively learned representations are already highly performant, and finetuning leads to a small degradation. DeepMicro, on the other hand, performs worse and does require finetuning in order to achieve improved classification performance.
>
> >6	What's the masking percentage (for Random Subcompositions) in the contrastive learning experiments?
>
> For each datapoint, the masking percentage is sampled from a U(0,1) (as per Section 3.2).
>
> > Minor changes
>
> - We have added several non-microbiome compositional datasets to our analysis and found our methods consistently performant (Section 4.3).
> - We have renamed Random Subcompositions to Compositional Feature Dropout.
> - Figure 3 was moved to the main text.
> - Limitations of our work are mentioned at the top of Section 3 and in Appendix D.
> - The remaining minor points are addressed directly in the updated manuscript.

---

> > ### Comment · Reviewer_ymr8 · 2022-08-03
> > **Good changes made; Contrastive learning section needs clarity improvements**
> >
> > I thank the authors for answering my questions and making changes.
> >
> > I continue to find the section on contrastive learning rather rusty, and the clarity can be well-improved. For example, the second and third paragraphs start almost identical:
> > - second paragraph starts "As a point of comparison, we use DeepMicro [42], a deep representation learning architecture tuned 279 specifically for microbiome data."
> > - third paragraph starts "To ensure a fair comparison, our contrastive model uses the same encoder architecture from DeepMicro".
> >
> > I suggest the authors to make the contrastive learning section more clear (note: I find the rest of the paper better written; consider moving some implementation details in the appendix).
> >
> > I suggest the first paragraph of the contrastive learning section specify that you show a *proof-of-concept* of contrastive representation learning to microbiome CoDa.
> >
> > I *trust* the authors will make my suggested changes with the "contrastive learning section".
> >
> > Consequently, I updated my rating to 7 "Accept".

---

> > > ### Author Response · Authors · 2022-08-04
> > > **Thank you**
> > >
> > > Many thanks for reviewing our updates and for your response. We agree that the contrastive learning section could be improved by contextualizing our experiments upfront; we shall explicitly state its goal as a proof-of-concept for contrastive learning on CoDa. We also agree with moving some of the implementation details to the Appendix, in order to focus on explaining the contrastive experiment more carefully, with additional clarifying remarks. Thank you again for the detailed feedback.

---

### Official Review · Reviewer_V2Ko · 2022-07-11

**Rating:** 6
**Confidence:** 3
**Soundness:** 3 good
**Presentation:** 3 good
**Contribution:** 3 good

**Summary:**

3 new data augmentation strategies are proposed for compositional data and these methods are tested on microbiome data. With the proposed methods, state-of-the-art results in disease prediction tasks including colorectal cancer, type 2 diabetes, and Crohn’s disease are reached. They show that the proposed data augmentations can also be used for representation learning in a contrastive setting similar to SimCLR.


**Questions:**

I do not have any questions.

**Ethics Review Area:**

["I don’t know"]

**Limitations:**

- I liked that you also reported the datasets, which your method could not achieve state-of-the-art, which might indicate transparency.

- My major concern is the one I listed in the Weaknesses as #1. I think the paper needs additional experiments using other types of CoDa for this venue as stated or at least it needs some rewriting in the claims.

**Strengths And Weaknesses:**

Strengths:

- In almost all of the cases the used augmentations improve the SOTA methods. This is a result of the fact that data augmentations can be used on top of the existing methods / can be orthogonally combined with existing methods.

- Paper is mostly written well.

- Related work seems to be well-studied.

- There is a lack of comprehensive study in the field of data augmentations for compositional data.

- I liked the namings of the proposed data augmentations.

Weaknesses:

- The proposed data augmentations could be used for other types of compositional data besides human microbiome data. In my opinion, performance on other types of CoDA should be tested and reported for a machine learning conference like NeurIPS. Even though the methods might not work with other types of compositional data, this could as well be added to limitations and appendix. More importantly, there is a tendency to claim in some parts of the paper that the proposed data augmentations are for CoDa, not only for microbiome data. Here is an example from the "Conclusion": "By combining ideas from data augmentation with the principles of CoDa, we have defined 3 novel augmentation strategies for CoDa: Aitchison Mixup, Random Subcompositions, and Compositional CutMix." In my opinion, if you are making such claims, it is necessary to provide results in other CoDa datasets.

- Adapting data augmentations to CoDa is the main novelty of the paper. However, in my opinion, adaptations are to some extent straightforward and not very novel. "CoDa analogue of Mixup", "standard preprocessing step in CoDa", and "CoDa analogue of CutMix"  already give some hints regarding the originality of the method. On the other hand, the science builds on top of each other, thus I could say this is not a big but a medium-level concern.

- In most cases, the increase in the SOTA performance seems not so big, even though the datasets are pretty small. I would expect a higher increase for meaningful data augmentations for such small datasets. However, considering that it is hard to come up with data augmentations for CoDa, this could also be a minor point.

---

> ### Author Response · Authors · 2022-08-02
> **Additional benchmarks on non-microbiome CoDa have been added; other comments have been addressed**
>
> Many thanks for the constructive feedback.
>
> >• The proposed data augmentations could be used for other types of compositional data besides human microbiome data. In my opinion, performance on other types of CoDA should be tested and reported for a machine learning conference like NeurIPS. [...]
> >
> >•	My major concern is the one I listed in the Weaknesses as #1. I think the paper needs additional experiments using other types of CoDa for this venue as stated or at least it needs some rewriting in the claims.
>
>
> We agree that our work would benefit from expanding our experiments to other common types of CoDa, a point that was also raised in other reviews.
> That said, we also note that the human microbiome remains one of the biggest and most high-impact areas of application for CoDa methodology.
>
>
> Thus, in addition to our 12 microbiome classification tasks, we have now added 8 separate non-microbiome compositional datasets to our benchmarks. These datasets are well-studied in the CoDa literature and were selected to represent a diverse mix of scientific applications that are relevant to CoDa practitioners, including materials science, hydrochemistry, geochemistry, metabolomics, oncology, and nutritional science.
>
> The full detail regarding these datasets and corresponding experimental results is provided in a new Subsection (see 4.3 in the updated manuscript).
> We highlight the following key results:
> - As was the case on our microbiome benchmark, we find that our data augmentations again deliver consistent improvements in classification performance across a large majority of our non-microbiome datasets and predictive models.
> - On 6 out of 8 non-microbiome datasets, the best performing model was one using Aitchison Mixup.
> - On these non-microbiome datasets, tree-based models strongly outperformed network models. Aitchison Mixup never hurt the performance of these tree-based models (and did so rarely for our network models).
>
>
> >• Adapting data augmentations to CoDa is the main novelty of the paper. However, in my opinion, adaptations are to some extent straightforward and not very novel. "CoDa analogue of Mixup", "standard preprocessing step in CoDa", and "CoDa analogue of CutMix" already give some hints regarding the originality of the method. On the other hand, the science builds on top of each other, thus I could say this is not a big but a medium-level concern.
>
> We agree that our method is straightforward to understand and implement; however we see this simplicity, combined with its consistent empirical performance, as a key strength of our work (note that another reviewer also commented positively on this feature).
> Arguably, much the same can be said about many key works on data augmentation. For example, Mixup [ref. 65 from the manuscript], CutMix [64], and RandAugment [12] could be portrayed as "straightforward" extensions of SMOTE [9], CutOut[14], and AutoAugment [11], respectively.
> Nevertheless, these works provided convincing empirical evidence for their methods and have become highly influential as a result.
> Our work aims to strike a similar balance between simple (and computationally cheap) methodology and careful empiricism, applied to a previously unexplored domain.
> Importantly, extending the success of data augmentation to this novel domain did require some nontrivial developments, such as incorporating the Aitchison geometry of the simplex.
>
>
> > •	In most cases, the increase in the SOTA performance seems not so big, even though the datasets are pretty small. I would expect a higher increase for meaningful data augmentations for such small datasets. However, considering that it is hard to come up with data augmentations for CoDa, this could also be a minor point.
>
> We note the following:
> - The performance gains hold consistently across datasets. This consistency, combined with the ease of implementation and low computational cost, makes a strong case for the use of our augmentations even for "small" improvements in performance.
> - In the context of high-stakes decisions such as microbiome-based medical diagnostic tests, even small percentage improvements in classification accuracy can have an outsized human impact (e.g., lives saved from early detection of colorectal cancer).
> Moreover, since our data augmentation techniques can be combined orthogonally with any classifier, as the predictive accuracy of these diagnostic tests continues to increase over time, we expect to consistently provide meaningful improvements to their performance.
> - The magnitude of the performance gains that we have obtained in our CoDa benchmarks is comparable to those reported in other domains [14,64,65].
> Note that, while computer vision benchmarks tend to have many more datapoints than our own, their dimensionality and model capacity are correspondingly larger, so one need not necessarily expect higher gains on our smaller datasets.

---

> > ### Comment · Reviewer_V2Ko · 2022-08-03
> > **I appreciate the new experimental results and your replies**
> >
> > Thank you for the reply.
> > - I appreciate the new experimental results on different types of datasets.
> > - I acknowledge that I understand the argument that you see your method as a simple one, and I agree. I also see your point on novelty: "For example, Mixup [ref. 65 from the manuscript], CutMix [64], and RandAugment [12] could be portrayed as "straightforward" extensions of SMOTE [9], CutOut[14], and AutoAugment [11], respectively. Nevertheless, these works provided convincing empirical evidence for their methods and have become highly influential as a result."
> > Again I agree with these sentences mostly. However, I still see the "often minor" increase in the accuracy as a concern. Mixup [ref. 65 from the manuscript], CutMix [64], and RandAugment[12]: in my opinion these methods also often present more impressive performance increase as well as they provide experiments in more diverse data modalities or computer vision (which is a very popular field and where a lot of research goes on), which contributes to being " highly influential" as you proposed in your reply.
> >
> > To sum up, I think additional experimental results contribute to the paper significantly and I only have minor concerns now. I will update my recommendation accordingly.

---

> > > ### Author Response · Authors · 2022-08-03
> > > **Thank you**
> > >
> > > Many thanks for reviewing our additional experiments and for the the kind response, much appreciated.

---

### Meta-Review · Area_Chair_kQAH · 2022-08-22

**Recommendation:** Accept
**Confidence:** Certain

**Metareview:**


This paper proposes three different, easy to implement methods for data-augmentation when learning models on compositional data (where each feature lies in a potentially high-dimensional simplex). The basic idea is that for such data, it is important to create augmentations that respect the fact that data are within the simplex. There was consensus among the reviewers that this work should be accepted. This work was simple, interesting and results in empirical improvements across various choices of models of outcomes. I think this work can have an impact for those building predictive models in applications of machine learning such as microbiome data.


**Award:**

No

---

### Decision · Program_Chairs · 2022-09-14

Accept